# Looking Into the Water by Unsupervised Learning of the Surface Shape

**Ori Lifschitz**
Hatter Department of Marine Technologies
Charney School of Marine Sciences, University of Haifa
Haifa, Israel
https://github.com/OriLifschitz/RDR-SuGrad

**Tali Treibitz**
Hatter Department of Marine Technologies
Charney School of Marine Sciences, University of Haifa
Haifa, Israel
ttreibitz@univ.haifa.ac.il

**Dan Rosenbaum**
Department of Computer Science
University of Haifa
Haifa, Israel
danro@cs.haifa.ac.il

## Abstract

We address the problem of looking into the water from the air, where we seek to remove image distortions caused by refractions at the water surface. Our approach is based on modeling the different water surface structures at various points in time, assuming the underlying image is constant. To this end, we propose a model that consists of two neural-field networks. The first network predicts the height of the water surface at each spatial position and time, and the second network predicts the image color at each position. Using both networks, we reconstruct the observed sequence of images and can therefore use unsupervised training. We show that using implicit neural representations with periodic activation functions (SIREN) leads to effective modeling of the surface height spatio-temporal signal and its derivative, as required for image reconstruction. Using both simulated and real data we show that our method outperforms the latest unsupervised image restoration approach. In addition, it provides an estimate of the water surface.

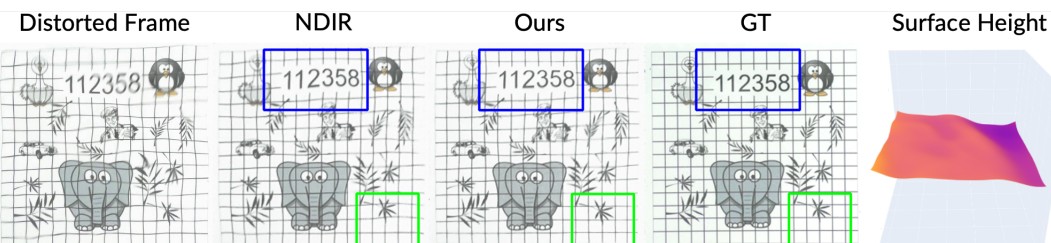

Figure 1: Surface waves distort the appearance of underwater scenes viewed from the air due to refractions following Snell's law as seen in the distorted frame on the left. Our method aims at reconstructing the original scene and performs better than NDIR [11]. For example, the numbers marked in blue and the grid marked in green. In addition, our method outputs the surface height. The height presented on the right corresponds to the distorted frame on the left. Example taken from the *elephant* dataset from [9].

39th Conference on Neural Information Processing Systems (NeurIPS 2025).

# 1 Introduction

Observing objects in the ocean from aerial platforms can significantly increase the observation scale and speed since underwater vehicle operation is complex and expensive. Therefore, it is used in a variety of scientific and operational applications. For example, measuring the scale of coral bleaching after a warming event or a storm [3]. However, the effects of refraction between the air and water can strongly distort the apparent position and shape of objects and features below the water's surface, hindering the observations. The distortion is directly related to the surface shape through its gradients, connecting the water surface with the underlying scene in a single image. Ocean wave measurements are very useful for coastal and ocean science and engineering and represent an active research field in physical oceanography [1, 15]. Measurement of sea surface height also informs studies of the sea surface microlayer [25].

Here, using a short input sequence of an underwater object seen from above, we aim to reconstruct its fine details and undistorted structure, as well as the surface height in each point. An example is shown in Fig. 1. Real waves are a superposition of several types of wave from various sources with different wave periods and amplitudes. Large-scale ground truth for this problem is only available by simulations that do not encompass this range of surface waves. Thus, it is difficult for supervised methods to generalize to real-world examples. However, distorted sequences contain a wealth of information on the constant underlying scene that can be leveraged in an unsupervised method.

We formulate the unsupervised learning signal of reconstructing the observed distorted images by first estimating the water surface height and then using the estimation and its spatial derivatives to compute the pixel distortion map. We implement this using a neural representation network based on periodic activation functions (SIREN [17]), which has proven to be effective and efficient in modeling continuous signals and their derivatives.

Our method outperforms previous unsupervised methods using a simpler training setup and additionally provides surface height estimation. We demonstrate this both on real-world and simulated data. All our code and data will be made available upon publication.

# 2 Related work

## 2.1 Imaging through turbulent water.

The problem of looking through water has attracted attention since the earlier days of computer vision [14] as it poses interesting and applicative physics-based challenges. Early works [5, 8, 24] were based on finding and stitching together distortion-free patches from the distorted image sequence. In [8] the authors formulate the reconstruction problem as a manifold embedding problem and propose a modified convex flows technique to robustly recover global distances on the manifold. In [5] the authors suggest a multistage clustering algorithm combined with frequency domain measurements. In [24] the authors propose to first find an ensemble of distortion free ("lucky") patches and then proceed to estimate the Fourier phase and the Fourier magnitude of the clean image.

A different line of techniques is based on model-based tracking to reconstruct the clean image. In [23] the authors propose building a spatial distortion model of the water surface using the wave equation. The model enables them to design a tracking technique tailored for water surfaces. Using their method, they were able to use a shorter sequence of 61 frames instead of the long sequences required by the lucky patches techniques *e.g.* 800 in [8] or 120 in [24].

A refracted image sequence contains strong physics cues on the underlying scene and the water surface. This was used by [9] who estimated optical flow between key feature points to estimate object trajectories within the sequence. Using a compressive sensing solver they used these trajectories to estimate the entire motion field and reconstruct the scenes. Alternatively, [26] used the fact that water refraction changes the viewpoint to develop a structure-from-motion like solver for an image sequence captured by a stationary camera. They are able to simultaneously retrieve the structure of the water surface and the static underwater scene geometry. Sulc et al. propose a parameter-free, Snell's-law–based objective for monocular reconstruction of an arbitrary refractive surface from a single distorted view given known background texture and geometry. Unlike our setting (unsupervised restoration from a short sequence), their method assumes a known background and directly optimizes surface geometry via a geometric ray-consistency error [18].

Supervised deep learning methods require a training dataset. In [12] a dataset was acquired using a computer monitor displaying images from ImageNet [4] placed under a transparent water tank with a pump to generate water surface movements. Their network consists of two parts, a warping net to remove geometric distortion and a color predictor net to further refine the restoration.

Thapa *et al.* [21] generated a synthetic dataset using the wave equation for three types of waves: ripple waves, ocean waves, and Gaussian waves. The dataset was then used to train the following network: three parallel CNNs that generalize features from each input (consecutive distorted frames), and then uses recurrent layers to refine the CNN-predicted distortion maps by enforcing the temporal consistency among them. Then, a GAN is used to predict the distortion-free image. FSRN [20] was trained to estimate the water surface based on a known reference background in the water.

Li *et al.* [11] present a two-stage unsupervised network. The first stage consists of a grid deformed for *every* input image that estimates the distortion field. Then, an image generator outputs the distortion-free image. The optimizer for generating the distortion-free image by minimizing pairwise differences between the captured input images, the network's predicted distorted images, and resampled distorted images from the distortion-free image. Our model is also based on unsupervised reconstruction, modeling the image and the distortions using different networks, however we use a single SIREN [17] network conditioned on time that outputs the surface height. Using the height we compute the offset of each pixel in order to reconstruct the observed images. The advantage is that the predicted distortion is grounded in the temporal process of moving waves, and in addition this enables a direct prediction of the water surface.

## 2.2 Neural fields

Representing data using *neural fields* has gained significant focus in recent years. These models, also known as *implicit neural representations* are used to model a continuous signal by predicting the value of the signal given a position in space as input. For example in images this corresponds to predicting pixel color given the 2D pixel position. This representation has proven effective for image compression [6], 3D modeling [13], PDE dynamics forcasting [27] and as a basic representation for different downstream tasks [7].

One implementation of neural fields that we adopt in this work is SIREN [17]. This model is an MLP with sinusoidal activation functions, which was proven to be both effective and efficient for image modeling, achieving high accuracy with smaller networks and faster training times. One advantage of using periodic activation functions, is that the signal and its derivatives become similar in nature. To demonstrate this Sitzmann *et al.* [17] train a model to predict a signal by supervising the training loss with the gradients of the signal. In our work we show that this arises naturally from the physical formulation of the problem, as the image distortion caused by the water surface is directly related to the spatial derivatives of the surface height.

# 3 Method

Fig. 2 summarizes our method. Given an arbitrary number of video frames taken from air, we train our model per sequence to reconstruct the distorted frames and can then use our model structure to remove the distortion caused by the water surface. This is achieved by modeling the underlying image and the water surface separately.

## 3.1 Assumptions

We assume a static planar underwater scene at an unknown depth $h_0$ below a water surface. The scene is fronto-parallel to an orthographic camera and the camera is held in air outside of water. The interface between the refractive medium (water) and air is dynamic, *i.e.*, wavy water-surface. Like [9, 16, 21, 22] we also assume small fluctuating water-waves, *i.e.*, the maximum surface fluctuation, $\max_{x,t} |h(x,t) - h_0|$, is small compared to the average water-height $h_0$. This is a reasonable assumption for many marine environments such as river beds and shallow coral reefs. If the camera exposure time is not sufficiently short in relation to the wave phase-velocity then the resulting image will suffer from motion-blur. While we do not make explicit assumptions regarding shutter speed, camera frame rate and the water-surface waves phase speed, our proposed method can handle the motion-blur commonly occurring in James *Real1* dataset [9] which is considered a

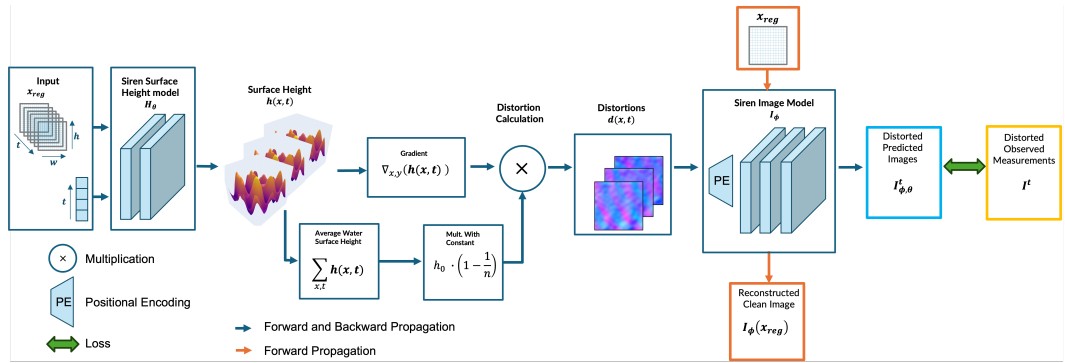

Figure 2: Our architecture. From the left, regularized 2D spatial grids $x_{\text{reg}}$ and time $t$ are inputs to a SIREN [17] network that outputs surface height per frame. The gradient of the output heights, along with its average across $t$, is used for calculating distortions as in Eq. 1. These are then used in another SIREN network to output the reconstructed image $I_\phi(x_{\text{reg}})$ and the distorted images $I^t_{\theta,\phi}$. The predicted distorted images and the observed distorted images $I^t$ are used in the loss calculation.

hard dataset for underwater refractive distortion removal due to large frame-to-frame motion and motion-blur. In Fig. 7 we present real-world results where our assumptions do not fully hold and show that our method is able to reconstruct a plausible underlying clean image, see Sec. 5 for further discussion.

## 3.2 Surface Gradient

By modeling the surface of the water above the image we can compute the refraction offset of every pixel based on Snell's law (illustrated in Fig. 3). According to Snell's law, under first-order approximation [22], the distortion function (warping) $d(x, t)$ can be related to the height of water surface $h(x, t)$ at the 2D spatial position $x$ and time $t$:

$$d(x, t) = \left(1 - \frac{1}{n}\right) h_0 \nabla h(x, t) \ , \tag{1}$$

where $h_0$ is the average water height above the scene, and $n$ is the relative refraction index between air and water.

## 3.3 Architecture

The architecture (Fig. 2) of our model consists of two parts. Both parts are implemented using SIREN models [17] which are used as neural fields to model both the water surface height at different times points $h(x, t)$, and the fixed underlying image.

**Surface height model $H_\theta$.** The first part of our model is a SIREN neural field that models the 2D surface height signal across different time points. This model takes as input a two dimensional position in the image space $x$, and one-dimensional point in time $t$, and predicts the surface height at that position and time $H_\theta(x, t)$. Using a SIREN architecture is well-suited for modeling distortions through surface height as it allows for the efficient prediction of a signal and its spatial derivatives simultaneously. We use a single SIREN to 1) predict the water surface height $h(x, t)$ which is used to compute $h_0$ by averaging the outputs across the 2D space $x$ and time $t$; and 2) predict $\nabla h(x, t)$ by computing the gradients of the output of the network with respect to its spatial input $x$. Given the computed $h_0$ and $\nabla h(x, t)$ we compute the pixel distortions $d(x, t)$ due to light refraction for each observed image at time $t$ using Eq. 1 and use it to reconstruct the observed images. We then compare this prediction with the observed images to compute the loss that we use to optimize the weights of the networks.

**Image model $I_\phi$.** This is implemented using a standard neural field SIREN where each pixel position is fed to the network along with a positional encoding using random Fourier features [19],

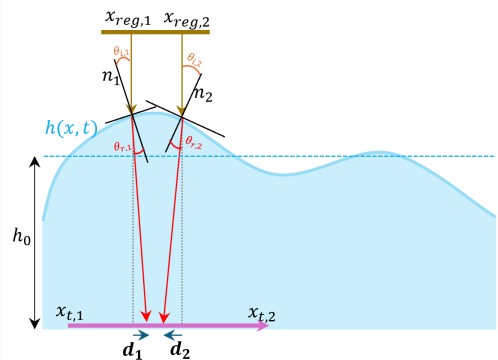

Figure 3: According to Snell's law light passing through an interface between media with different refraction indices changes its angle (refracts). Thus, when an orthographic camera views an object submerged in water from air, the object changes its geometrical appearance as a function of the normals to the surface.

and is trained to predict the pixel values $I_\phi(x)$. Feeding this model with a set pixel positions on the image's regular grid $x_{\text{reg}}$ will output the set of pixel values for the distortion-free image $I_\phi(x_{\text{reg}})$, and feeding the model with a set of distorted pixel positions $x_t = x_{\text{reg}} + d(x_{\text{reg}}, t)$ will result in a distorted image which we denote by $I_{\theta,\phi}^t$, since it is generated using both models $H_\theta$ and $I_\phi$ and therefore depends on both sets of weights. To formulate this explicitly,

$$I_{\theta,\phi}^t = I_\phi \left( x_{\text{reg}} + d \left( H_\theta(x_{\text{reg}}, t) \right) \right) \quad , \tag{2}$$

where $d(H(\cdot))$ corresponds to using Eq. 1 on the height prediction.

### 3.4 Training

Given a set of distorted frames across time $I_t$, we follow the training paradigm in [11] consisting of two stages. The first training stage can be seen as an initialization of the weights, by training the height network $H_\theta$ to output a height that corresponds to zero distortion for all pixels, and training the image network $I_\phi$ to predict the average distorted image. This is implemented using the loss ($|\cdot|$ represents the $L_1$ norm):

$$\mathcal{L}_{\text{init}}(\theta, \phi) = |d\left(H_\theta(x_{\text{reg}}, t)\right)| + \sum_t |I_\phi(x_{\text{reg}}) - I_t| \quad . \tag{3}$$

In the second stage of training the loss is computed by the reconstruction of all the distorted images. The loss is given by:

$$\mathcal{L}(\theta, \phi) = \sum_t \left| I_{\theta,\phi}^t - I_t \right| \tag{4}$$

This is a significant simplification compared to [11] which required 3 different loss terms for training. We note that both the reconstructed and observed images inherently involve the gradient of the surface height signal computed through Eq. 1 for the reconstruction and through the physical process of refraction in the observed image. Therefore this loss forms a real world application of the ability of SIREN to model a signal by supervising it with the derivatives.

## 4  Experiments

To test our method we use three datasets. We use the James *Real1* dataset [9], which contains 7 sequences of images acquired in a water tank using 50 fps acquisition rate (examples in Fig. 4). Additionally, we use the *TianSet*, which is also a real captured dataset by Tian and Narasimhan [23] using a 125 fps camera. We note that *Real1* is considered a more challenging dataset as it has larger frame-to-frame motion and includes motion blur (due to water-waves). We also generate a *synthetic* dataset using the method in [20], with 3 wave types, resulting in 11 sequences of images. We compare our method to NDIR [11], which is our unsupervised baseline and to Li *et al.* [12] which is the

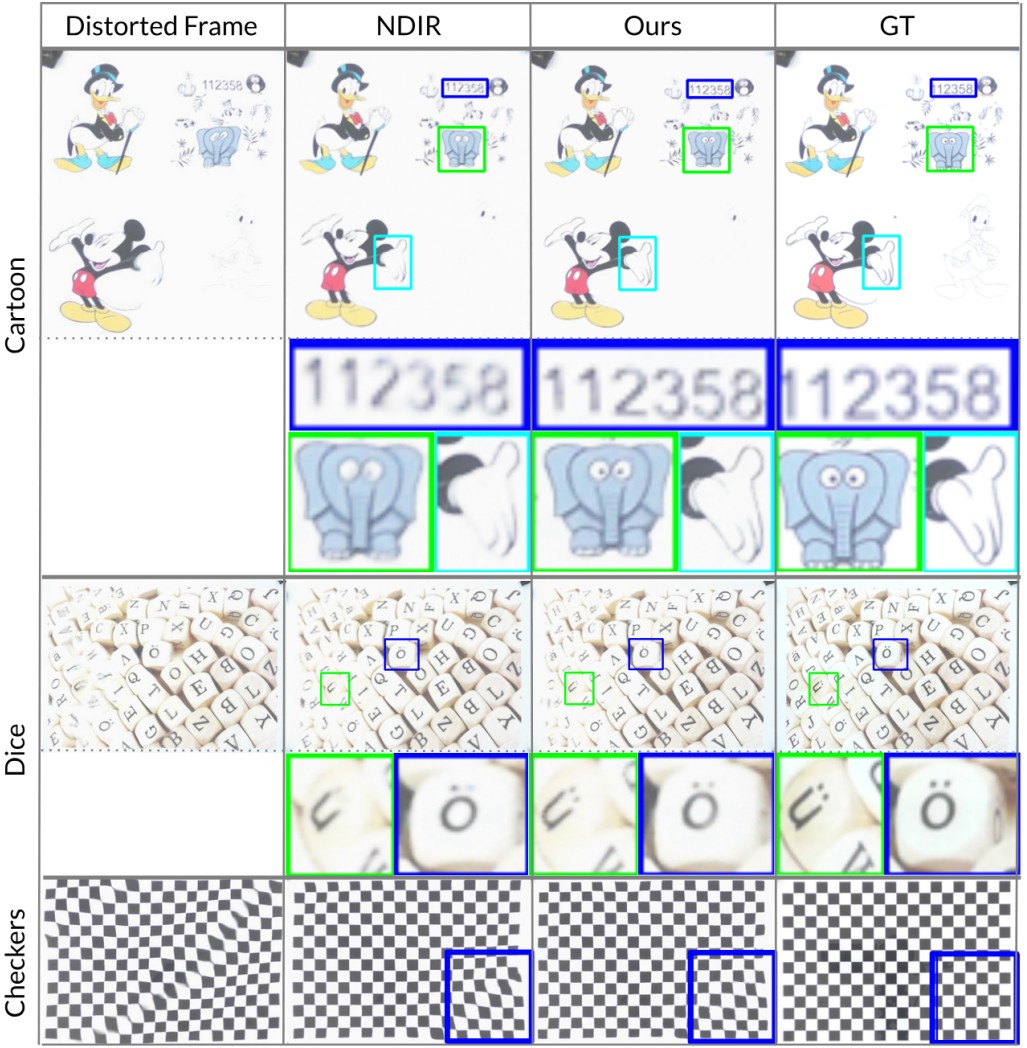

Figure 4: Results on the *Real1* dataset [9]. Marked squares indicate areas where our results are sharper than the baseline. Note sharper details in the *cartoon* and *dice* sequences, as well as straighter squares in the *checkers* sequence in our method.

state-of-the-art supervised method on single images. In all experiments we use a sequence size of 10 frames (except for the batch-size ablation). We conduct ablation studies to validate benefits of modeling surface-height and spatio-temporal information, to examine design choices and to evaluate the impact of the input sequence size (Sec. 4.3). Additional results and videos are provided in the supplementary material.

**Implementation details.** In all experiments we use a 2-layer network for $H_\theta$ and a 3-layer network for $I_\phi$, both trained with the Adam optimizer. The input to $I_\phi$ is augmented with random Fourier positional encoding with the bandwidth factor set to 8. We use two sets of hyperparameters. One set for both real datasets (*Real1* and *TianSet*) and the other for the synthetic dataset. More details on the configurations of the hyperparameters, the amount of iterations on each training stage, hardware setup, memory usage, runtime, and reproducibility scripts are provided in the supplementary material (*"Implementation Details"*). Our implementation is based on the code of [11].

## 4.1 Image restoration

A qualitative comparison of our results on the *Real1* dataset is shown in Figs. 1 and 4 where green and blue rectangles mark areas of interest. In the *cartoon* sequence our method better aligns several

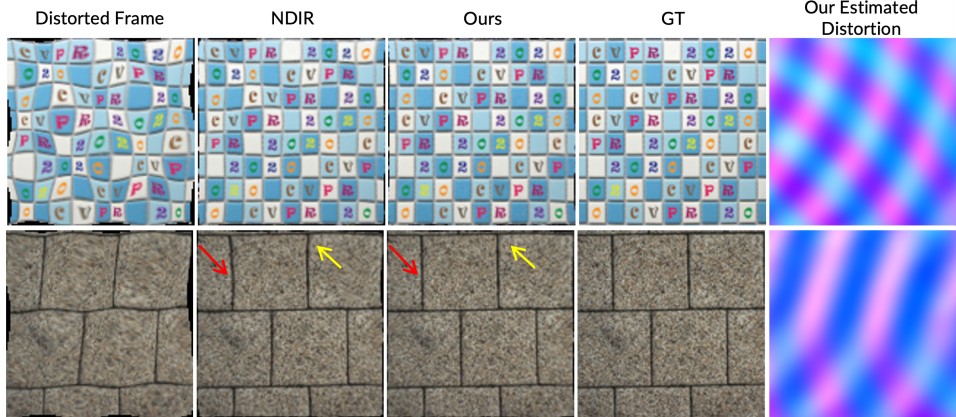

| Distorted Frame | NDIR | Ours | GT | Our Estimated Distortion |

Figure 5: Examples from the simulated dataset following [20]. Our results have less distortions, manifesting in straighter lines. The estimated distortions match the apparent distortions in the acquired frames.

Table 1: Quantitative comparison results on the Real1 dataset from [9] and TianSet [23]. We compare our method with NDIR [11], Li *et al*. [12], and NeRT [10]. Our method achieves the best overall performance on Real1, with the best SSIM and LPIPS across both datasets. It outperforms prior methods on most individual sequences and metrics, demonstrating consistent superiority. Minor exceptions include NDIR achieving slightly higher PSNR and SSIM on *Math*, Li *et al*. obtaining the best SSIM in isolated TianSet cases, and NeRT occasionally strong in PSNR and SSIM but substantially worse LPIPS.

| | | NDIR [11] | | | Li *et al*. [12] | | | NeRT [10] | | | Ours | | |
|---|---|---|---|---|---|---|---|---|---|---|---|---|---|
| | Dataset - Sequence | PSNR ↑ | SSIM ↑ | LPIPS ↓ | PSNR ↑ | SSIM ↑ | LPIPS ↓ | PSNR ↑ | SSIM ↑ | LPIPS ↓ | PSNR ↑ | SSIM ↑ | LPIPS ↓ |
| Real1 JamesSet [9] | Bricks | 20.83 | 0.55 | 0.21 | 19.71 | 0.55 | 0.18 | 20.48 | 0.68 | 0.35 | **21.34** | **0.59** | **0.16** |
| | Cartoon | 21.86 | 0.77 | 0.16 | 18.75 | 0.71 | 0.19 | 21.63 | **0.84** | 0.37 | **22.37** | 0.79 | **0.12** |
| | Checker | 14.10 | 0.55 | 0.12 | 12.36 | 0.45 | 0.26 | 16.03 | 0.72 | **0.09** | **14.27** | 0.58 | 0.10 |
| | Dices | 18.50 | 0.51 | 0.11 | 16.23 | 0.41 | 0.24 | 18.27 | 0.59 | 0.21 | **19.15** | **0.57** | **0.09** |
| | Elephant | 15.63 | 0.31 | 0.19 | 14.63 | 0.23 | 0.26 | 15.91 | 0.41 | 0.38 | **15.95** | **0.33** | **0.17** |
| | Eye | 21.10 | 0.82 | **0.10** | 18.36 | 0.78 | 0.12 | 15.75 | 0.51 | 0.21 | **21.42** | **0.83** | **0.10** |
| | Math | 24.07 | **0.62** | **0.11** | 19.92 | 0.55 | 0.13 | 23.34 | 0.55 | 0.39 | 23.98 | 0.60 | **0.11** |
| | **Average** | 19.44 | 0.59 | 0.14 | 17.14 | 0.53 | 0.20 | 18.77 | 0.61 | 0.29 | **19.78** | **0.61** | **0.12** |
| TianSet [23] | Small | 19.67 | 0.33 | 0.26 | 18.42 | 0.34 | **0.22** | 19.82 | **0.37** | 0.33 | **19.90** | 0.36 | **0.22** |
| | Middle | 16.76 | 0.41 | 0.15 | 15.82 | **0.45** | 0.21 | **17.37** | 0.44 | 0.25 | 17.10 | 0.43 | **0.13** |
| | **Average** | 18.22 | 0.37 | 0.20 | 17.12 | 0.39 | 0.22 | **18.60** | **0.40** | 0.29 | 18.50 | **0.40** | **0.17** |

areas, e.g., the numbers, elephant, hand, and as a result reconstruct more details in these drawings. In the *dice* sequence our method compensates better for the distortion and as a result reveals more fine details in the letters. In the *checkers* sequence the motion of the squares in the bottom right corner is better compensated. Fig. 5 displays results from two simulated scenes where our result shows straighter lines and less distortions. The estimated distortion matches the apparent distortions in the acquired frame.

Tab. 1 summarizes quantitative results for all three methods, ours and [9, 11], on the real datasets [9, 23]. Our approach achieves the highest overall metrics. Detailed per-sequence evaluations are also presented along with the standard deviation for each sequence. In *Eye* and *Math* our method is on-par with NDIR [11]. In all other sequences in both datasets, our method achieves the best LPIPS, while also providing surface-height prediction. Our method achieves the best PSNR and SSIM values, with only one exception on the *Math* sequence. We note that PSNR often favors overly smooth reconstructions and is thus less indicative of perceptual quality, whereas LPIPS aligns more closely with sharpness and fidelity, as can be seen in the qualitative comparison (Fig. 4). Finally, Tab. 2 shows results on the simulated dataset, where our method is superior in all metrics compared to the baseline method on every wave-type.

Table 2: Quantitative comparison results on a simulated dataset following [20] on three wave types. Our method performs better than NDIR [11], as can also be seen in Fig. 5.

| Wave type | Method | PSNR ↑ | SSIM ↑ | LPIPS ↓ |
|---|---|---|---|---|
| Gaussian | NDIR [11] | 20.70 | 0.64 | 0.12 |
| | Ours | **20.92** | **0.65** | **0.11** |
| Ripple | NDIR [11] | 21.50 | 0.846 | 0.07 |
| | Ours | **23.07** | **0.92** | **0.05** |
| Ocean | NDIR [11] | 15.52 | 0.54 | 0.15 |
| | Ours | **16.07** | **0.58** | **0.14** |

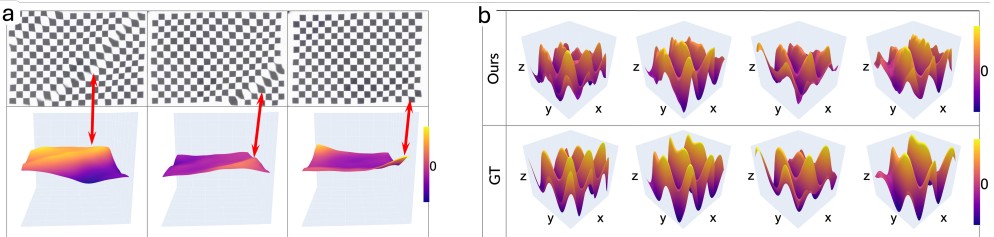

Figure 6: Examples of surface height reconstructions. a) Checkers set from the *Real1* dataset [9]. The strong curvatures in the surface match the strong distortions in the input image. b) A ripple wave from the simulated dataset based on [20]. We show reconstructions of four frames evolving with time, where our reconstruction closely matches the ground truth used for simulation.

## 4.2 Water surface estimation

In addition to reconstructing the underlying image, our method estimates the water surface height of every frame in the sequence. Examples are shown qualitatively on sequences from the *Real1* dataset [9] in Figs. 1, 6a. The estimated surface curves match large distortions and blurs in the input image. For a quantitative comparison, we use the estimated depth in the *Synthetic* dataset. The average root mean square errors (RMSE) and absolute relative errors (Abs Rel) are 0.115 and 0.0635, respectively. These results are on par with previous methods shown in the supplementary of [26]. Fig. 6b shows surface height estimations on a *ripple* sequence evolving with time. We see that our estimations closely follow the ground-truth height shape over time.

## 4.3 Ablations.

All ablations were performed using the more challenging *Real1* dataset. The quantitative results presented in Tables 3 and 4 are the average over the entire dataset.

**Modeling surface-height and spatio-temporal information.** We compare our final architecture with two ablations of our method (Tab. 3). In **ablation 1**, we predict per-frame pixel distortions $x_t$ using separate networks for each time step instead of a single, time-conditioned network. **Ablation 2** employs a shared network across time while still predicting $x_t$, ignoring surface height. Both ablations use SIRENs for sub-networks and neglect refraction modeling, precluding surface height estimation. The key distinction is that **ablation 2** incorporates temporal conditioning. Our final proposed method (**Our method**) integrates surface-height prediction and temporal conditioning, achieving the best PSNR, LPIPS and SSIM, aligning well with qualitative results.

**Design choices.** We conduct ablation studies to validate our design choices (Tab. 3). In NDIR [11] the authors use in the second stage of training a loss function composed of three terms which are based on two different ways to compute predictions of the distorted images. The first way is computing $I_{\theta,\phi}^t$ using Eq. 2, and the second way is to generate the regular image $I_\phi(x_{\mathrm{reg}})$ and then use the distortion to warp the image $W_{\theta,\phi}^t = \mathrm{warp}\left(I_\phi(x_{\mathrm{reg}}), d\left(H_\theta(x_{\mathrm{reg}}, t)\right)\right)$. Using these two predictions and the observed image, they compute the loss by comparing the three possible pairs for each observed frame

Table 3: Design-choice ablations. We examine the loss terms. *No init* skips the first training stage (initialization). *No positional encoding* replaces the positional encoding module with another SIREN layer as suggested by [2]. Ablations 1 and 2 are discussed in 4.3. The results are the average results on *Real1* dataset.

| Metric | $L1$ Our Method | $L2$ | $L3$ | $L1 + L2 + L3$ | $L1$ No init | $L1$ No positional encoding | Ablation 1 | Ablation 2 |
|---|---|---|---|---|---|---|---|---|
| PSNR ↑ | **19.78** | 18.79 | 15.32 | 19.42 | 19.42 | 19.07 | 19.42 | 19.47 |
| SSIM ↑ | **0.613** | 0.54 | 0.49 | 0.60 | 0.58 | 0.57 | 0.59 | 0.59 |
| LPIPS ↓ | **0.121** | 0.15 | 0.77 | **0.13** | 0.19 | 0.25 | 0.18 | 0.18 |

Table 4: Comparison of different batch sizes. Presented are the average results on *Real1*.

| Batch size | PSNR ↑ | | SSIM ↑ | | LPIPS ↓ | |
|---|---|---|---|---|---|---|
| | NDIR | Ours | NDIR | Ours | NDIR | Ours |
| 5 | **17.81** | 17.69 | 0.54 | **0.54** | 0.17 | **0.14** |
| 6 | 17.90 | **18.56** | 0.55 | **0.55** | 0.16 | **0.15** |
| 7 | 18.04 | **18.87** | 0.56 | **0.57** | 0.16 | **0.15** |
| 8 | 18.15 | **18.98** | 0.56 | **0.57** | 0.15 | **0.15** |
| 9 | 18.18 | **19.15** | 0.57 | **0.58** | 0.16 | **0.15** |
| 10 | 19.14 | **19.78** | 0.58 | **0.613** | 0.14 | **0.121** |

in time:

$$\mathcal{L}(\theta, \phi) = \sum_t \left| I_{\theta,\phi}^t - I_t \right| + \left| W_{\theta,\phi}^t - I_t \right| + \left| I_{\theta,\phi}^t - W_{\theta,\phi}^t \right| \tag{5}$$

We test different combinations of these 3 loss terms, and find that for our model, using only the first term as formulated in Eq. 4 results in better performance. This shows that our model can attain better results while using a simplified training setup compared to NDIR, for which it was shown that all 3 loss terms are required for training stability. We also test against training without the initialization phase and without positional encodings and find them beneficial.

**Input sequence size.** We perform batch size ablations (Tab. 4), showing our method consistently achieves the best LPIPS and SSIM across all tested sizes. For PSNR, our approach also outperforms NDIR [11] starting from six frames. Interestingly, for a batch size of five, the competing method achieves higher PSNR but 0.03 worse LPIPS. Given LPIPS' stronger correlation with perceptual quality, even a 0.03 difference is often visually noticeable.

## 4.4 A Real-World Scene with Partial Fulfillment of Assumptions

We test our method in a real-world setting (Fig. 7) without an orthographic camera and with a non-fronto-parallel scene due to complex object geometry. Motion blur is present (*e.g.*, green rectangle), yet our method reconstructs both the planar grid-like structure and non-planar corals, demonstrating robustness to assumption violations.

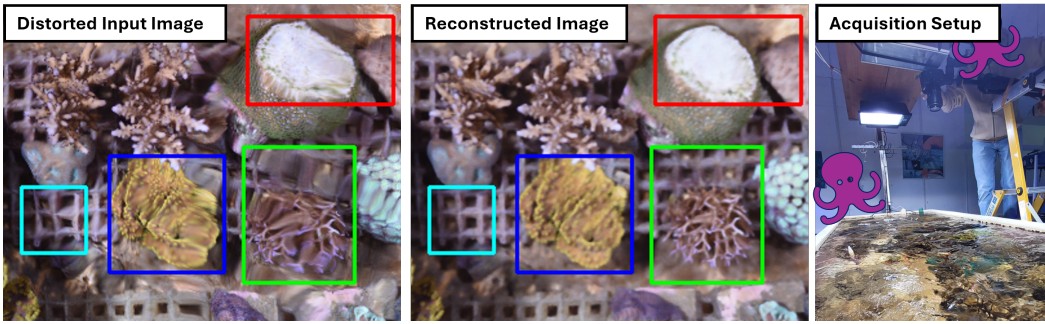

Figure 7: **Real-world scene in a coral tank**, in which distortions in 3D objects are corrected. Some loss of detail stems from limited network resolution.

# 5 Summary

Looking into the water from air is an interesting physics-based problem where image distortion is tied to the surface shape through Snell's law. Since the water surface is changing temporally, a sequence of images acquired from the same viewpoint provides different distorted views of the same scene. Our method leverages this information to reconstruct a single image of the undistorted scene.

Real-world acquisition of such scenes and their ground-truth is very challenging. Several methods simulated datasets using different formulations of wave equations. Nevertheless, in the real-world the actual waves are a super position of different wave-forms with multiple amplitudes and periods, and also depend on the bottom type. Thus, there is strong advantage for developing unsupervised methods that leverage the physical cues from the image sequences and do not rely on pre-training.

We present an unsupervised method that both reconstructs the underlying distorted scene and the surface shape in each temporal frame. With the exploding popularity of aerial drones our method has numerous applications in ocean surveys, fish farm monitoring, and also drowning detection both in the ocean and in swimming pools. Future work includes evaluating our restoration as a pre-processing step for downstream feature matching and monocular SLAM [28].

Beyond these positive impacts, potential negative implications may include misuse for unauthorized surveillance in underwater settings or misinterpretation of reconstructions in safety-critical scenarios. Our work is primarily foundational and aimed at environmental, marine research and safety applications. Such concerns highlight the need for responsible deployment and further investigation into failure modes.

**Acknowledgments.** The research was funded by Israel Science Foundation grant #1951/23, Israeli Ministry of Science and Technology grants #1001577600 & #1001593851, EU Horizon 2020 research and innovation programme GA 101094924 (ANERIS), and the Maurice Hatter Foundation.

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
