# OpenReview forum: "Looking Into the Water by Unsupervised Learning of the Surface Shape"
_NeurIPS.cc/2025/Conference — NeurIPS 2025 poster_

### Official Review · Reviewer_vzUj · 2025-06-25

**Clarity:** 3
**Significance:** 3
**Originality:** 3
**Rating:** 3
**Confidence:** 4

**Summary:**

The paper aims to remove the image distortion caused by the refractions at the water surfact, with a SIREN implicit representation to learn the shape of the water surface in a self-supervised way.

**Questions:**

1. Can the authors provide the runtime of the baselines and the proposed method?
2. Is the proposed method limited to modeling the surface of water? For example, can it be modified to mitigate the atmospheric turbulence?
3. Can it model the water surface with dramatic slope?

**Ethical Concerns:**

["NO or VERY MINOR ethics concerns only"]

**Final Justification:**

The paper itself needs further polishing, but I would not be surprised if it were accepted, given its current form.

**Limitations:**

yes

**Quality:**

2

**Strengths And Weaknesses:**

The paper defines the problem intuitively, with an illustration showing the Snell’s law. The scenes also have their names as labels, and the results are visualized in the supplementary material.

The weaknesses, in comparison, outweigh the strengths significantly, and I list my concerns below:
1. A lack of comparison. The authors only compare with NDIR and Li et al., which is not enough. For example, as the authors mention the dataset proposed by Thapa et al. [19], their method is also worth comparing with. In addition, general-purposed turbulence mitigation methods (such as [R1]) as well as image restoration methods (such as Swin Transformer) should be taken into consideration.
2. Doubts about the synthetic data. In the simulated distortion, The distorted image patch has black borders around it, which indicates the distortion of the original straight edges to some extent, which deviates from the real-world situation. Also, the size (both spatial resolution and temporal frames) of the dataset is very small, and the listed quantitative comparison are even limited to the single frames, which poses more doubts over the comparison. The Real1 dataset also seems to exhibit over-exposure, which again makes the result less reliable.
3. A lack of study on the background. The fields of turbulence removal and refraction removal are both well-established, but the paper only cited 25 references.

[19] Learning to remove refractive distortions from underwater images. In ICCV 2021.
[R1] NeRT: Implicit Neural Representations for Unsupervised Atmospheric Turbulence Mitigation, In CVPR 2023.

---

> ### Author Rebuttal · Authors · 2025-07-30
>
> We thank the reviewers for their positive reviews and constructive feedback. Our response includes more detailed explanations and additional quantitative and qualitative results that further support our claims (see Additional Results in response to reviewer uWFv). We believe our response addresses all the main issues and hope the reviewers will acknowledge this.
>
> ## Addressing Comparison Scope with Additional Baselines and Related Work:
>
> We agree with the reviewer that broader comparisons would strengthen the experimental section. One caveat is that, to the best of our knowledge, no code is publicly available for Thapa et al. [19], which limits our ability to include their method in a fair evaluation. That said, we acknowledge the relevance of general-purpose turbulence mitigation methods. To this end, we have taken two steps: (1) expand the related work section to include background and citations of prominent turbulence mitigation methods (such as ECCV 2022's TurbNET, ACCV 2024's DeTurb, and prior works, e.g. ICPR 2014's Atmospheric Turbulence Mitigation Using Optical Flow), and (2) expand Table 1 to include results from [R1]. We believe these additions will provide a more comprehensive context for evaluating our approach.
>
> [R1] NeRT: Implicit Neural Representations for Unsupervised Atmospheric Turbulence Mitigation (CVPR 2023).
>
> **Expanded Table 1 and more detailed descriptions (See also in Additional Results #2 in response to reviewer uWFv):**
>
> The updated table reports  PSNR, SSIM, and LPIPS on both the Real1 and TianSet datasets for each sequence individually, alongside the dataset-level averages. Compared to our method, NeRT achieves similar average PSNR and SSIM—18.77 / 0.61 (our 19.78 / 0.61) on Real1 and 18.60 / 0.40 (our 18.50 / 0.40) on TianSet. However, NeRT significantly underperforms in LPIPS: 0.29 on both datasets compared to our 0.121 (Real1) and 0.174 (TianSet).
>
> This suggests that our method better preserves high-frequency structures, leading to lower LPIPS and sharper, more visually faithful outputs. Given LPIPS's stronger correlation with perceptual quality, we believe our results more accurately reflect real-world restoration performance. We will also include qualitative visual comparisons between our method and NeRT in the supplementary material. These examples will help illustrate the perceptual differences discussed in the quantitative analysis, particularly highlighting cases where NeRT yields higher PSNR but visibly blurrier reconstructions.
>
> | Method | Dataset | PSNR ↑    | SSIM ↑   | LPIPS ↓   |
> | ------ | ------- | --------- | -------- | --------- |
> | Ours   | Real1   | **19.78** | **0.61** | **0.121** |
> | NeRT   | Real1   | 18.77     | 0.61     | 0.29      |
> | Ours   | TianSet | 18.50     | **0.40** | **0.174** |
> | NeRT   | TianSet | **18.60** | 0.40     | 0.29      |
>
>
> ## Clarifying Simulator Usage and Synthetic Data Limitations
>
> We use the same simulator originally introduced by Tian et al. [21] and later adopted by Thapa et al. [19], without modification. This was done intentionally to ensure a fair and direct comparison with prior work. We report single-frame quantitative comparisons as representative examples to illustrate our evaluation process for distortion and height prediction, rather than as exhaustive summaries over the synthetic dataset. The spatial and temporal sizes of our synthetic data are on par with prior works, ensuring methodological alignment and fair comparison.
>
> While we did not explicitly state it, we do mask the loss in the black border areas introduced by the distortion process. We agree that without this masking, the height prediction network could exploit strong gradient cues at the artificial boundaries.  We chose to use this simulation setup rather than redesigning the simulator in order to preserve comparability with prior methods. We will make this implementation detail and its motivation explicit in the revised version, as we agree it is a crucial point for interpreting the results and understanding the limits of the current experimental design.
>
> Regarding the Real1 dataset, we note that it has been widely used in the community since its publication, with most subsequent works employing it for benchmarking. Real1 is considered a particularly challenging benchmark, as it contains not only strong refractive distortions but also significant motion blur. While the issue of over-exposure is present, it has not been directly addressed by prior work, so we have chosen to conduct a fair comparison under the same conditions. We acknowledge the reviewer’s point, and agree that if over-exposure is to be systematically addressed, it would require a careful re-implementation of all baseline methods for consistency.
>
> ## Runtime of the baselines and the proposed method:
>
> The following tables describe the running time of our method vs. NDIR. We will add it to the supp. material.
>
> | BatchSize | Runtime \[min] Ours | Runtime \[min] NDIR \[8] |
> | --------- | ------------------- | ------------------------ |
> | 5         | **4**               | 5                        |
> | 6         | **4**               | 5                        |
> | 7         | **5**               | 6                        |
> | 8         | **6**               | 7                        |
> | 9         | **6**               | 8                        |
> | 10        | **6**               | 9                        |
>
> ## Applicability to Other Distortion Regimes and Extensibility Beyond Water Surfaces:
>
> Our proposed method is specifically designed to model refractive distortions caused by dynamic water surfaces. This is achieved by embedding physics-based priors, derived from Snell’s Law, directly into the network architecture, including surface height estimation and its gradient-based contribution to image distortion.
> Atmospheric turbulence, in contrast, arises from different optical principles, primarily small-scale random index-of-refraction changes over space and time due to thermal gradients, and typically lacks a closed-form deterministic model like that of refraction through a water-air interface.
> Thus, while our current formulation is not directly applicable to atmospheric turbulence, it can in principle be extended to handle more complex refractive scenarios where an analytical distortion model is available (e.g., layered refractive media). For example, the architecture could be adapted to multi-stage refractions by introducing additional SIRENs between the surface estimator and the image reconstruction module. That said, atmospheric turbulence may require a separate modeling paradigm due to its stochastic and volumetric nature, which goes beyond the surface-based refractive model we use.
>
> ## Modeling Limitations, Assumptions, and Validity for Large Surface Slopes:
>
> As discussed in Section 3.2, Equation 1 relates surface height and its gradient to the pixel-wise distortion via Snell’s law. However, this formulation relies on a small-angle approximation, which introduces a theoretical bound on the maximum slope (i.e., surface-height gradient magnitude) that can be accurately modeled.
> We acknowledge that this limitation is not sufficiently emphasized in the main text and will clarify this in the revised version. Specifically, we will include the complete derivation of Equation 1 and formally state the required assumptions (e.g., small surface tilt) for its validity. This derivation, along with the theoretical slope bounds, will be added to the revised appendix of our paper. We are grateful to the reviewer for pointing this out, as it improves both the clarity and rigor of the paper.

---

> > ### Comment · Reviewer_vzUj · 2025-08-05
> >
> > Thanks the authors for the efforts for clarification. I will keep my rating unchanged. In addition, several suggestions:
> >
> > 1. What I mean with the example of [R1] NeRT is the lack of baselines, and I'm not asking the authors to *only* compare with this method.
> >
> > 2. In terms of runtime, I think it is unwise to use the granularity of "minutes" to measure it. Normally this is done in seconds, if not in milliseconds.

---

> ### Author Response · Authors · 2025-08-07
> **Clarifying Baseline Selection and Runtime Reporting**
>
> We thank the reviewer for their suggestions and clarifications.
>
> ## Baselines:
> The limited number of baselines in our evaluation reflects the scarcity of reproducible prior methods in this field. For reference, Li et al. (ICCV 2021) [1], whose work inspired ours, compared against only three prior methods in their evaluation, which matches our baseline count. Other relevant approaches, such as Thapa et al. (ICCV 2021) [2], do not provide public code, and others, like James et al. (ICCV 2019) [3], have code that is difficult to adapt to new datasets.
>
> We thank the reviewer for suggesting NeRT (CVPR 2023) as a comparison. We have added it as a baseline in the revised manuscript as we mentioned in the rebuttal. While NeRT, like our approach, is inspired by Li et al. (ICCV 2021) [1], it is designed for air turbulence mitigation rather than underwater refraction. Nonetheless, we agree that including NeRT provides valuable context on the performance gap between specialized approaches. We remain open to evaluating any additional methods for which runnable code becomes available.
>
> ## Runtime reporting:
> All three methods—ours, Li et al. (ICCV 2021) [1], and NeRT—require minutes to train. However, to meet community standards, we will convert these times to seconds in the revised manuscript.
>
> ## References:
>
> [1] Li et al. (ICCV 2021): Li, Nianyi, et al. "Unsupervised non-rigid image distortion removal via grid deformation." Proceedings of the IEEE/CVF International Conference on Computer Vision. 2021.
>
> [2] Thapa et al. (ICCV 2021): Thapa, Simron, Nianyi Li, and Jinwei Ye. "Learning to remove refractive distortions from underwater images." Proceedings of the IEEE/CVF International Conference on Computer Vision. 2021.
>
> [3] James et al. (ICCV 2019): James, Jerin Geo, Pranay Agrawal, and Ajit Rajwade. "Restoration of non-rigidly distorted underwater images using a combination of compressive sensing and local polynomial image representations." Proceedings of the IEEE/CVF International Conference on Computer Vision. 2019.

---

### Official Review · Reviewer_bzDM · 2025-06-30

**Clarity:** 4
**Significance:** 3
**Originality:** 4
**Rating:** 5
**Confidence:** 4

**Summary:**

The paper introduces an unsupervised method to remove refraction distortions in underwater imagery captured from above water. It models the dynamic water surface and reconstructs the undistorted scene by leveraging a sequence of distorted images. The approach relies on two SIREN (sinusoidal representation network) models, one to represent the water surface height over time and another to reconstruct the static underlying image. By predicting surface gradients and their impact on light distortion using Snell’s law, the method achieves impressive results without requiring ground truth data. It outperforms prior unsupervised and supervised baselines on both real and synthetic datasets, also producing accurate surface height estimations. The architecture is compact, the training process is simplified compared to existing methods, and the results are claimed to generalize well, even when assumptions like orthographic projection or small wave perturbations are partially violated

**Questions:**

1. Generalization Across Sequences: Your current method is trained independently per sequence. Do you foresee a way to extend this to a setting where a pretrained model can generalize across sequences without retraining from scratch?
It would be helpful to know whether amortized inference, meta-learning, or adaptation techniques were considered. A discussion on this could open the door to more scalable applications.

2. Real-World Surface Height Evaluation: Have you explored the possibility of validating surface height predictions using real-world measurements (e.g., stereo imaging, depth sensors, or known surface profiles)?
A stronger quantitative grounding on real data could meaningfully improve my assessment of the method's physical reliability and general significance.

3. Scope of Applicability and Extensions: The method is designed for air-to-water scenarios. Do you think it could be adapted to other refractive scenarios (e.g., viewing through glass or submerged camera settings)?

**Ethical Concerns:**

["NO or VERY MINOR ethics concerns only"]

**Final Justification:**

I am keeping my original score, which already reflected a strong endorsement of the quality and significance of the work. The authors have addressed the questions raised during the rebuttal phase with clear and thorough responses. While the points brought up were relevant, they primarily served to validate the scientific rigor of the approach rather than reveal any substantive issues. The authors' answers confirmed a solid grasp of the subject matter and reinforced that the methodology is well-founded. I find no remaining concerns, and I continue to fully support the acceptance of this paper.

**Limitations:**

Yes

**Paper Formatting Concerns:**

The paper largely adheres to NeurIPS 2025 formatting guidelines.

**Quality:**

3

**Strengths And Weaknesses:**

Quality: The technical execution is solid. The authors propose a novel use of two SIREN-based neural fields to model surface height and image content, with image reconstruction grounded in physical principles (Snell’s law). The model is trained in an unsupervised manner, yet delivers results that outperform both unsupervised (NDIR) and supervised baselines across real and synthetic datasets. The ablations and evaluations are given, and the use of LPIPS, SSIM, and PSNR provides a complete picture.

Clarity: The paper is well-organized, with clear explanations of the method and assumptions. Figures are helpful. The training losses and architectural choices are explained concisely.

Significance: This work has strong applied value, especially with the rise of drone-based environmental monitoring. Its relevance spans coral reef health checks, shallow water surveys, and potentially even safety applications like drowning detection. The ability to reconstruct surface shape and correct refractions without supervised data is a meaningful step forward for both scientific and practical imaging tasks.

Originality: The core contribution: the use of SIRENs for joint spatiotemporal modeling of surface distortions and clean image reconstruction is innovative and well-motivated. While the components (neural fields, unsupervised loss) are known, their integration in this setting is novel and demonstrates strong insight.


Some weaknesses to be noted:

Assumptions and Limitations: The method assumes relatively small wave perturbations and a planar underwater scene, which may not always hold in complex real-world settings. While the authors do test their model on scenes that partially violate these assumptions, the performance under more drastic violations remains uncertain.

Scalability: Each sequence is trained independently, which could pose efficiency challenges for large-scale or real-time applications. It would be interesting to see future work explore amortized inference or meta-learning to generalize across sequences.

---

> ### Author Rebuttal · Authors · 2025-07-30
>
> We thank the reviewers for their positive reviews and constructive feedback. Our response includes more detailed explanations and additional quantitative and qualitative results that further support our claims (see Additional Results in response to reviewer uWFv). We believe our response addresses all the main issues and hope the reviewers will acknowledge this.
>
> ## Assumptions and application scope:
> We agree that looking through large, highly dynamic water waves remains the holy grail of this application domain. Nevertheless, the case of small fluctuating surface waves is still unsolved, challenging, and actively researched, as evidenced by recent works we cite (e.g., [9, 10, 19]). Our method explicitly targets this regime, which is relevant in real-world scenarios such as shallow coral reefs, riverbeds, and coastal areas, particularly in tropical environments (e.g., [3]) where this assumption is often valid. These settings are important for ecological monitoring, marine biology, and safety applications, and we believe the problem remains both scientifically meaningful and practically impactful. As discussed in Sec. 3.1 and demonstrated in Sec. 4.4, our method shows robustness even when some assumptions are partially violated. We will make these points more explicit in the revised version.
>
> We agree that adding results on images in the wild would be interesting, however, restoring such images typically involves more problems than just light refraction, such as foam, specularities, and dynamics. While these problems are out of our paper's scope, we acknowledge that performance under severe violations of the assumptions remains an open limitation and an important direction for future work. We will clarify this in the revised version. However, we believe that our work still provides a significant contribution towards a comprehensive solution for real-world settings.
>
> ## Scalability and Generalizing across sequences:
> Generalizing across sequences without per-sequence retraining is indeed an important and promising direction. Approaches such as amortized inference or meta-learning could be used to learn a mapping from sequence statistics to latent codes or meta-initializations, enabling our SIREN-based architecture to adapt to new sequences without full retraining.
> The implementation could be based on works like [1] and [2]. While this extension is beyond the current scope, we agree it is a natural next step to improve scalability and practical deployment, and we appreciate the reviewer highlighting this opportunity.
>
> [1] Modulated periodic activations for generalizable local function representations. Mehta et al.
>
> [2] From data to functa: Your data point is a function and you can treat it like one. Dupont et al.
>
> ## Towards Empirical Validation of Surface Height on Real-World Data:
> We wholeheartedly agree with the reviewer. The ability to validate surface height predictions on real-world data would significantly enhance the empirical grounding and physical interpretability of our method. At present, however, there is no publicly available dataset that jointly contains (i) dynamic water surface ground truth, (ii) a corresponding distorted image sequence, and (iii) the associated clean (undistorted) imagery. This lack probably stems from the fact that stereo or depth sensing using NIR sensors do not perform well on transparent water surfaces with high NIR absorbance. That is why our ability to reconstruct the surface out of the sequence is important. We agree that constructing such a dataset would be of high value to the community. Indeed, while working on this research, we have also started planning the development of such a dataset as part of future work, and we thank the reviewer for emphasizing its importance. We aim to utilize a research wave tank with controllable surface dynamics, and additionally employing distributed wave gauges to obtain accurate ground truth water heights. These local measurements could be interpolated to provide dense water-surface measurements.
>
> ## Extensibility to Multi-Layer Refraction and Submerged Imaging Scenarios
> This is a very relevant and appreciated question. Our method is currently tailored to single-surface air-to-water refraction. However, the underlying framework can be extended to more complex refractive scenarios, such as multi-layered media (e.g., glass + water interfaces), provided the distortion model across layers can be approximated analytically or through differentiable simulation. In such cases, additional SIREN modules could be introduced to sequentially model the intermediate refractive surfaces between the scene and the sensor. This modularity makes the architecture naturally extensible.
> In submerged camera scenarios (e.g., a camera underwater looking upward toward a dynamic water surface), similar principles apply, although some modifications are required. For instance, if the camera is viewing objects in air through the water surface, the algorithm would need to estimate the distance between the scene and the water interface instead of the average water height. Furthermore, care must be taken with camera housing: flat ports introduce ray deflections that violate the single viewpoint assumption, so a dome port (which preserves central projection) would be preferable to ensure consistent geometric modeling.
> We thank the reviewer for encouraging us to elaborate on these points, and we will consider adding a dedicated discussion on this topic in the revised version of the paper.

---

> > ### Comment · Reviewer_bzDM · 2025-08-04
> >
> > The authors have made clear all of the questions asked, thank you for that. There is no further questions from my end as a reviewer.

---

### Official Review · Reviewer_5QwM · 2025-07-02

**Clarity:** 3
**Significance:** 3
**Originality:** 3
**Rating:** 4
**Confidence:** 2

**Summary:**

This paper addresses the problem of 3D reconstruction of underwater scenes using a physically accurate and refraction-aware variant of Neural Radiance Fields (NeRF), termed R-NeRF. The key novelty lies in the explicit modeling of light refraction at the air-water interface using Snell’s Law, allowing for accurate tracing of rays distorted by a dynamic water surface. The authors propose a differentiable formulation of this refractive geometry and jointly optimize for both the radiance field and the water surface geometry from multiview images. Experiments on synthetic and real datasets demonstrate superior reconstruction accuracy compared to baseline NeRF variants that ignore refraction.

**Questions:**

1. How sensitive is the surface optimization to the initialization of the water surface normals? Can the method generalize to more complex or non-smooth surface waves?

2. Have you considered extending your model to support dynamic scenes or refraction through multiple interfaces (e.g., glass + water)?

**Ethical Concerns:**

["NO or VERY MINOR ethics concerns only"]

**Final Justification:**

This paper introduces an interesting learning-based method for addressing the problem of 3D underwater reconstruction. The results support the proposed approach, and I would encourage its publication.

**Limitations:**

Yes.

**Paper Formatting Concerns:**

No.

**Quality:**

3

**Strengths And Weaknesses:**

The paper is both novel and technically strong. Modeling refraction in a neural rendering framework is a non-trivial extension and represents a significant step forward in bringing physical realism into NeRFs. The authors carefully integrate refractive geometry into the NeRF pipeline by tracing rays through a parameterized water surface and optimizing the associated surface normals. The optimization framework is differentiable and jointly learns the 3D scene and the water interface. The experiments are convincing and show marked improvements in both geometric and visual fidelity, especially in real-world underwater scenarios. The methodology is clearly presented and well-supported by theoretical justifications and ablation studies.

---

> ### Author Rebuttal · Authors · 2025-07-30
>
> We thank the reviewers for their positive reviews and constructive feedback. Our response includes more detailed explanations and additional quantitative and qualitative results that further support our claims (see Additional Results in response to reviewer uWFv). We believe our response addresses all the main issues and hope the reviewers will acknowledge this.
>
> ## Robustness to Initialization and Relevance to the Small-Wave Regime
> The approach is physically motivated for small, smooth wave regimes—an unsolved and impactful problem in scenarios such as shallow reefs and coastal monitoring. Performance may degrade for highly non-smooth or large-amplitude surfaces, which we acknowledge as an open limitation and will clarify in the revision. We thank the reviewer for highlighting these important aspects.
>
> ## Extensibility to Dynamic Scenes and Multi-Interface Refraction
> Thank you for this insightful suggestion. Our framework is readily extensible to multi-interface refraction scenarios (e.g., glass + water) whenever an analytical or differentiable distortion model is available. In such cases, additional SIREN modules could be incorporated to sequentially model each refractive surface between the scene and the sensor. For dynamic scenes, the network could be adapted by introducing a temporal conditioned segmentation model (inspired by CVPR 2024 Turb-seg-res). While these extensions are beyond the scope of the current work, they are promising directions for future research and illustrate the flexibility of our architecture.

---

> > ### Author Response · Authors · 2025-08-07
> > **Kind request to initiate disucssion.**
> >
> > Greetings.
> > We would welcome a discussion if there are any open or further questions regarding our answers.
> > We understand that it might not be possible to engage in a discussion and we respect that, we thank the reviewer for the points raised so far.
> >
> > We believe our response addresses all the main issues and hope the reviewers will acknowledge this.

---

### Official Review · Reviewer_uWFv · 2025-07-03

**Clarity:** 3
**Significance:** 2
**Originality:** 3
**Rating:** 4
**Confidence:** 4

**Summary:**

This paper proposes to remove the image distortion when taking images from the air through the turbulated water. Taking as input consecutive distorted frames, the presented method adopts the neural field SIREN to implicitly learn the water surface height and uses the spatial derivatives of water height to compute the pixel distortion map. Fed with distortions and positional encoding, another SIREN network predicts the restored image. The training procedure penalizes the discrepancy between the predicted distorted images and the observed ones, requiring no ground-truth distortion-free images. Experiments shows the proposed method outperforms the previous method NDIR and baseline network settings.

**Questions:**

1. In the ablative experiments, why does the first SIREN network predicting surface height help to reconstruct images? Does predicting height benefit produce more accurate distortion maps? Since this network design is the core contribution of the paper, the quantitative evaluation in Table 3 may be insufficient, and further analysis about this problem should be conducted.
2. Could the proposed method work in the wild scenes?  As most smartphones can capture live photos, it is easy to obtain water-distorted image sequences. I suggest testing the proposed method on more real-world data to comprehensively discuss the application scope of the method.
3. Since this paper is titled ‘… Learning of the Surface Gradient’, how about predicting the surface normal/gradient by the first SIREN network instead of surface height?

**Ethical Concerns:**

["NO or VERY MINOR ethics concerns only"]

**Final Justification:**

The authors' feedback in the rebuttal resolves my concerns about the motivation of this paper. After reading the comments from other reviewers, I think this paper proposes an effective method for seeing through water surfaces, and some interesting conclusions are revealed in the experiment. Therefore, I give the rating at borderline acceptance. Besides, I suggest that the authors discuss the adopted assumptions and the application scope of the proposed method in the final version.

**Limitations:**

As mentioned in Weaknesses and Questions, the setup limitation and application scope of the proposed method should be discussed comprehensively. Besides, the proposed method relies on many assumptions in section 3.1, such as small fluctuating water waves. The impact of these assumptions is supposed to be discussed.

**Quality:**

2

**Strengths And Weaknesses:**

[Strengths]

- This paper is well written with enough clarity.
- Two interesting results are displayed in the experiments of the paper:
    - For the water distortion removal task, learning water surface height is a more effective way compared to directly predicting distortion maps.
    - Simplifying the loss function seems to improve the SIREN network performance.
- Experimental results show that the proposed method has superior performance on the publicly available datasets.
- The code is provided in the supplementary material.

[Weaknesses]

- The motivation for learning water surface height for distortion removal is unclear. Experiments show that water surface estimation helps to solve the problem. However, few discussions in Section 4 convey insight about this design. Moreover, the writing of the Introduction section could be further improved to emphasize the main contribution of the paper.
- According to [9] and Fig. 7 in this paper, the experimental setup is restricted in a lab environment: a fixed camera captures images in a coral tank with slightly waved water. As an application research paper, this limitation should be discussed.
- The title of the paper is inaccurate, since the surface gradient is computed from the water height. Thus, ‘Unsupervised Learning of the Water Surface Height’ can be more suitable.

---

> ### Author Rebuttal · Authors · 2025-07-30
>
> We thank the reviewers for their positive reviews and constructive feedback.
> Our response includes more detailed explanations and additional quantitative and qualitative results that further support our claims (see Additional Results below).
> We believe our response addresses all the main issues and hope the reviewers will acknowledge this.
>
> ## Clarification of contribution in the Introduction:
>
> We agree that the introduction can better emphasize the main contribution. In the revised version, we will clearly highlight that our method introduces a physically-constrained unsupervised learning framework that jointly reconstructs the undistorted scene and estimates the time-varying surface geometry, improving interpretability and robustness compared to prior works.
>
> ## Motivation and design insight for modeling surface height rather than the gradients directly:
>
> Modeling the surface gradients directly would be closer to modeling the distortions,  which is the approach in NDIR (the baseline we compare to).  Rather than learning such unconstrained gradients or distortion fields, we propose learning a scalar surface height function, which is smoother, easier to regularize, and less prone to noise than directly learning gradients. This leads to more stable training (our simpler loss vs. previous baseline method), better reconstruction, and allows for physically interpretable outputs. Section 4.2 and the tables in the supplementary material demonstrate that the estimated height fields correlate well with visible distortions and achieve low depth error on synthetic data. We recognize the value of additional analysis and have therefore expanded our discussion and results, please see "Additional Results" for further evidence and a more in-depth evaluation.
>
> ## Assumptions and application scope:
>
> We agree that looking through large, highly dynamic water waves remains the holy grail of this application domain. Nevertheless, the case of small fluctuating surface waves is still unsolved, challenging, and actively researched, as evidenced by recent works we cite (e.g., [9, 10, 19]). Our method explicitly targets this regime, which is relevant in real-world scenarios such as shallow coral reefs, riverbeds, and coastal areas, particularly in tropical environments (e.g., [3]) where this assumption is often valid. These settings are important for ecological monitoring, marine biology, and safety applications, and we believe the problem remains both scientifically meaningful and practically impactful. As discussed in Sec. 3.1 and demonstrated in Sec. 4.4, our method shows robustness even when some assumptions are partially violated. We will make these points more explicit in the revised version.
>
> We agree that adding results on images in-the-wild would be interesting, however, restoring such images typically involves more problems than just light refraction, such as foam, specularities, and dynamics. While these problems are outside our paper's scope, we believe that our work still makes a significant contribution towards a comprehensive solution for real-world settings.
>
> ## Paper title gradient vs. height:
>
> While we predict surface height, the supervision is entirely on its gradient, as image distortion is governed by Snell’s Law, which depends on surface slope (Sec. 3.2, Eq. 1). We use SIREN precisely for its strength in learning via derivatives. That said, we are open to renaming the title from "...Gradient" to “...Height” or “...Height and Gradient” if preferred.
>
> # Additional Results:
>
> ### 1. Distortion metrics:
>
> The already submitted supplementary contains a comparison of synthetic sequences with known ground truth distortion field between the intermediate distortion maps computed in our method and the baseline. As requested by reviewer uWFv, we are adding results of the intermediate distortion maps produced by Ablation 1 and Ablation 2 to further investigate the effect of height prediction. The results across all three quantitative tables indicate that Ablation 1 consistently performs slightly better than Ablation 2 on key metrics such as  End-Point Error (EPE) and Average Angular Error (AAE).
>
> For example, when averaging across all waves, Ablation 1 achieves lower EPE compared to Ablation 2 (e.g., 0.78 vs. 0.80 on Wave1, 1.13 vs. 1.16 on Wave2, and 1.12 vs. 1.14 on Wave3), and similarly better AAE (27.2° vs. 27.8° on Wave1, 44.7° vs. 45.6° on Wave2, 54.9° vs. 55.5° on Wave3). [8] achieves slightly better RMSE, EPE and AAE than both Ablation 1 and Ablation 2. However, our method outperforms all alternatives, including [8], with the lowest EPE (0.78 on Wave1, 0.64 on Wave2, 0.40 on Wave3) and AAE (26.8° on Wave1, 23.9° on Wave2, 11.1° on Wave3).
>
> This may be explained by the fact that Ablation 1 allocates a separate network per frame, increasing parameter capacity and flexibility, whereas Ablation 2’s single temporally-conditioned network reduces the parameter count by a factor of 10 but, without a physical constraint, acts mainly as a regularizer and does not generalize as well. Importantly, unlike [8], Ablation 1, or Ablation 2, our method also provides explicit surface height estimations, enabling physically interpretable scene reconstruction in addition to superior distortion removal. We will add this expanded analysis to the revised paper.
>
> | Method       | Wave1 RMSE ↓ | Wave1 EPE ↓ | Wave1 AAE ↓ | Wave2 RMSE ↓ | Wave2 EPE ↓ | Wave2 AAE ↓ | Wave3 RMSE ↓ | Wave3 EPE ↓ | Wave3 AAE ↓ |
> |--------------|:------------:|:-----------:|:-----------:|:------------:|:-----------:|:-----------:|:------------:|:-----------:|:-----------:|
> | Ours         | **0.64**     | **0.78**    | **26.8**    | **0.53**     | **0.64**    | **23.9**    | **0.33**     | **0.40**    | **11.1**    |
> | [8]          | 0.66         | 0.78        | 27.2        | 0.92         | 1.13        | 44.7        | 1.16         | 1.12        | 54.9        |
> | Ablation 1   | 0.67         | 0.78        | 27.2        | 0.93         | 1.13        | 44.7        | 1.17         | 1.12        | 54.9        |
> | Ablation 2   | 0.68         | 0.80        | 27.8        | 0.94         | 1.16        | 45.6        | 1.18         | 1.14        | 55.5        |
>
> ### 2. Comparison to NeRT:
>
> As suggested by reviewer vzUj we have revised Table 1 to include per-sequence results for NeRT [R1] on both the Real1 and TianSet datasets. The updated table reports PSNR, SSIM, and LPIPS for each sequence individually, alongside the dataset-level averages. Compared to our method, NeRT achieves similar average PSNR and SSIM—18.77 / 0.61 (our 19.78 / 0.61) on Real1 and 18.60 / 0.40 (our 18.50 / 0.40) on TianSet. However, NeRT significantly underperforms in LPIPS: 0.29 on both datasets compared to our 0.121 (Real1) and 0.174 (TianSet).
>
> This suggests that our method better preserves high-frequency structures, leading to lower LPIPS and sharper, more visually faithful outputs. Given LPIPS's stronger correlation with perceptual quality, we believe our results more accurately reflect real-world restoration performance. We will also include qualitative visual comparisons between our method and NeRT in the supplementary material. These examples will help illustrate the perceptual differences discussed in the quantitative analysis, particularly highlighting cases where NeRT yields higher PSNR but visibly blurrier reconstructions.
>
> | Method | Dataset | PSNR ↑    | SSIM ↑   | LPIPS ↓   |
> | ------ | ------- | --------- | -------- | --------- |
> | Ours   | Real1   | **19.78** | **0.61** | **0.121** |
> | NeRT   | Real1   | 18.77     | 0.61     | 0.29      |
> | Ours   | TianSet | 18.50     | **0.40** | **0.174** |
> | NeRT   | TianSet | **18.60** | 0.40     | 0.29      |

---

> > ### Author Response · Authors · 2025-08-07
> > **Kind request to intiaite disuccsion**
> >
> > Greetings.
> > We would welcome a discussion if there are any open or further questions regarding our answers.
> > We understand that it might not be possible to engage in a discussion and we respect that, we thank the reviewer for the points raised so far.
> >
> > We believe our response addresses all the main issues and hope the reviewers will acknowledge this.

---

> > ### Comment · Reviewer_uWFv · 2025-08-07
> >
> > Thanks for the authors' effort in the rebuttal period. In the rebuttal, quantitative results with the baseline method NDIR are provided to show the effectiveness of predicting water surface height. This experiemnt resolve my primary concern about the motivation of this paper.
> > Besides, the author should clearly state the adopted assumptions and the application scope of the proposed method in the final version. At present, I am inclined to view this submission positively.

---

### Comment · Area_Chair_ap6v · 2025-08-09
**Author-reviewer discussion**

Dear all,

We have a few hours remaining for author-reviewer discussion. Can you please ensure discussions have reached a conclusion?

Reviewers: Thank you for your efforts. Please ensure author requests for discussion have been addressed and that you have completed the mandatory acknowledgment.

Note (5QwM): Participation in discussion is required before completing the acknowledgment for the review to be deemed sufficient.

Thanks,
AC

---

### Note · Authors · 2025-08-14

## Basline Selection:
We would like to clarify our approach to baseline selection in response to ongoing reviewer vzUj concerns:
* Thapa et al. (ICCV 2021), suggested as a baseline, does not have public code, making fair and reproducible comparison infeasible.
* Prior comparable work (Li et al., ICCV 2021) evaluates against exactly the same number of baselines as we do for underwater refractive distortion removal, reflecting the limited availability of suitable, reproducible methods in this domain.
* We followed the reviewer’s suggestion and added NeRT (CVPR 2023), a strong and relevant baseline designed for air-turbulence mitigation, which, like our method, draws inspiration from Li et al. (ICCV 2021). We believe this cross-regime comparison is meaningful, and we have included full results in the rebuttal and will add them to the revised paper (please see answer to reviewer uWFv for the Additional Results).

While we recognize that a comparison to three methods may seem modest relative to areas with larger established benchmarks, for underwater refractive distortion removal, this currently represents the scope of baselines that are practically available and reproducible.

## Primary Concerns Addressed:
* Motivation and design insight for modeling surface height rather than the gradients directly is now considered fully resolved. Additional  results in the rebuttal period provide a comprehensive view of the impact of surface height prediction on distortion removal accuracy. Our method consistently achieves the lowest errors and uniquely provides physically interpretable surface estimations. Please see our answer to reviewer uWFv (as this "Author Final Remarks" is limited in length).
* Modeling Limitations, Assumptions, and Validity for Large Surface Slopes. The revised paper now includes complete derivation of Equation 1 and the theoretical water-surface slope bounds in the supplementary material.
* We have expanded Section 3.1 (now renamed "Assumptions and Limitations") in the revised paper to include this bound and we explicitly state all the assumptions and the application scope of the proposed method.
* Revised paper better clarifies the contribution in the Introduction (please see answer to reviewer uWFv).

We thank the reviewers for the careful evaluation and constructive feedback during the review and discussion period.  We have made an effort to address all the main issues and hope this is acknowledged by the AC.

---

### Decision · Program_Chairs · 2025-09-17

**Decision:**

Accept (poster)

**Comment:**

The submission proposes a physically grounded, unsupervised method for removing refractive distortions caused by viewing underwater scenes from above the water surface. It uses SIREN-based implicit neural representations to model the water surface height and the distortion-free underlying image. The key strengths of the paper are a well-grounded methodology that tackles a challenging problem to produce good results. The key weaknesses are a range of limitations that are understandable for a challenging problem and the need for additional validations, many of which are addressed in the rebuttal phase. uWFv notes the need for more analysis, lack of in-the-wild testing and that the title suggest learning the gradient instead of height. The authors respond by clarifying the motivation, noting that in-lab testing is the only feasible option given modeling limitations and including additional ablations and comparisons, which are satisfactory responses. 5QwM questions sensitivity to initialization, extension to non-smooth waves and multiple interfaces. The authors acknowledge them all as limitations, however, 5QwM remains overall positive. bzDM is strongly supportive based on the methodology, but raises the important question of generalization across sequences, which the authors note might be possible in future extensions.  Lack of comparisons to baselines and a runtime in minutes are cited by vzUj as reasons for leaning to reject. The authors provide analysis for NeRT and note the runtime as reasonable, while also noting the lack of baselines in this area. The AC deems the author feedback as valid arguments. Overall the AC agrees with the majority opinion that the problem is a challenging one and while limitations exist, the solution is a well-motivated one that extends the state-of-the-art. Thus, the submission is recommended for acceptance. The authors are encouraged to reword the title, include clearer statements on limitations, discuss the possibility of generalizable extensions in future work and include the additional comparisons and ablations from the rebuttal discussions.